# Vegetation communities on commercial developments are heterogenous and determined by development and landscaping decisions, not socioeconomics

Karen Dyson[1,2]*

**1** Research and Design for Integrated Ecology, Seattle, Washington, United States of America, **2** Urban Ecology Research Laboratory, University of Washington, Seattle, Washington, United States of America

* karenldyson@gmail.com

**Data Availability Statement:** Data and code can be found on GitHub: https://github.com/kdyson/Commercial-Vegetation-Paper, DOI: 10.5281/zenodo.3378131.

## Abstract

In urban ecosystems, woody vegetation communities and the ecosystem functions and habitat they provide are largely controlled by humans. These communities are assembled during development, landscaping, and maintenance processes according to decisions made by human actors. While vegetation communities on residential land uses are increasingly well studied, these efforts generally have not extended to other land uses, including commercial property. To fill this gap, I surveyed tree and shrub communities on office developments located in Redmond and Bellevue, Washington, USA, and explored whether aggregated neighborhood and parcel scale socio-economic variables or variables describing the outcome of development and landscaping actions better explained variation in vegetation communities. I found that both tree and shrub communities on office developments are heterogenous, with sites characterized by native or ornamental vegetation. The heterogeneity I observed in vegetation communities within one land use suggests that different ecosystem functions, habitat quality, and habitat quantities are provided on office developments. Greater provision of e.g. native conifer habitat is possible using currently existing developments as models. Additionally, the outcome of development and landscaping decisions explained more variation in community composition than the socio-economic factors found significant on residential property. Together with previous research showing that residential property owner attitudes and actions are more important than socio-economic descriptors, my results suggest that individual motivators, including intended audience, may be the primary determinant of urban vegetation communities. Future urban ecology research should consider sampling the vegetation gradient within land uses, better understanding individual motivation for vegetation management, and creating models of the urban ecosystems that account for alternate decision pathways on different land uses.

**Funding:** The author(s) received no specific funding for this work.

## Introduction

Woody vegetation community composition, structure, and distribution are largely controlled by human decisions and actions in urban ecosystems [1–7]. Development, landscaping, and ongoing maintenance are important milestones for management decisions that determine vegetation community characteristics. Changes to the vegetation community alter ecosystem service provision and habitat quality and quantity [1,8,9].

Development has replaced fire as the primary disturbance driver and precursor to new forest stands in the coastal Puget Sound region of Washington [4,7,10,11]. The mechanisms of disturbance when clearing and grading land for development include removing vegetation, removing topsoil, and compacting soil with heavy equipment [12–16]. Decisions made by developers and landowners at the time of development determine the extent of disturbance and influence future site conditions (Fig 1). For example, choosing to preserve existing trees determines legacy vegetation and influences stand characteristics like age and size [13].

Vegetation succession in urban ecosystems is determined through ecological processes such as dispersal and regeneration from seed banks and through landscaping and ongoing maintenance decisions made by developers and landowners [17]. The latter has become the dominant process [1,13,15,18–21]. Ornamental introduced shrubs, trees, or grasses are often chosen for landscaping, though using native species is becoming more common [1,14,19,22–24]. Once planted, these require significant ongoing maintenance inputs to arrest succession and maintain the desired aesthetic [1,17,25,26]. Along with trees retained through tree preservation policies, landscape plantings represent a significant portion of the vegetation on site and of the habitat quality and quantity available to other organisms [1,2].

Drivers determining vegetation management decisions, actions, and outcomes are multi-scalar, and include policy, community social pressures, aggregated neighborhood socio-economic status, and the motivations and preferences of individual landowners [27]. Relevant public policies include clearing and grading permitting processes, impervious surface maximums and minimums via parking space requirements, tree protection policies, canopy cover goals, and vegetation planting policies [28–30]. These policies are frequently enacted to protect ecosystem services, including carbon sequestration and aesthetic preferences [13,20,31–34].

Community drivers include the social norms and customs that influence individual behavior [27]. On residential properties, homeowners alter preferences for their own yards in response to the choices of nearby neighbor's yards [35], though their assumptions about

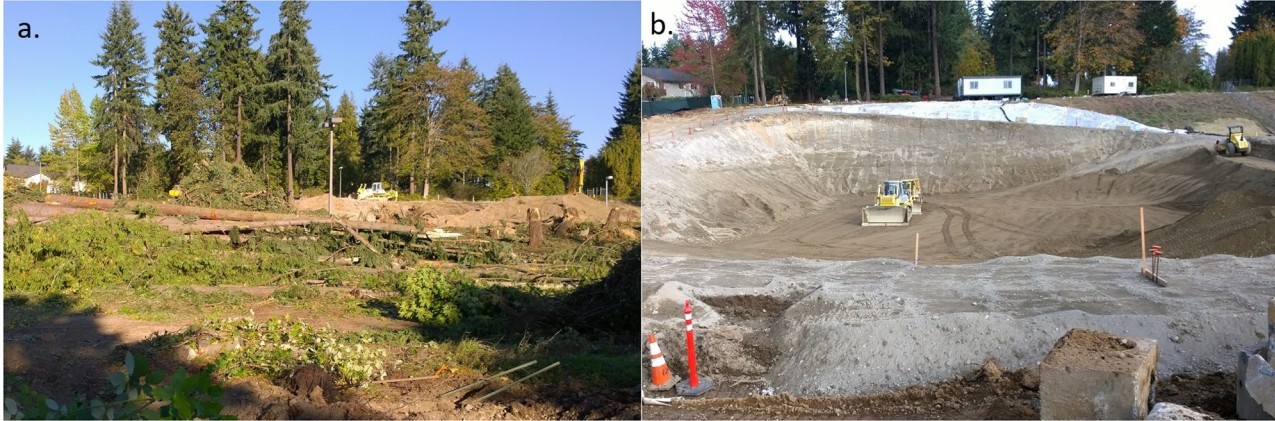

**Fig 1. Commercial development project located in Redmond, Washington.** Depicted: a. clearing the site of vegetation and b. grading the site and digging the foundation. Photo credit: K. Dyson.

neighbor preference are not always accurate [36]. On commercial properties, owners may alter preferences to appeal to prospective and existing tenants [37,38].

Neighborhood socio-economic status is often identified an important predictor of vegetation communities in studies of residential property. These variables are aggregated to the neighborhood scale, though they reflect group membership of the individual. Group membership often serves as a proxy for commonly held attitudes and ability to manipulate their environment [39], however they are also inextricably linked with systematic forces of inequality influencing the spatial distribution of wealth in a city [40]. Socio-economic variables correlated with canopy cover and other vegetation metrics include: current and historic household income [2,39,41–49], education level [21,45,48], ethnic composition [39,41,48,50], home value [51], home ownership [39], and housing age [2,44,46,47,52]. However, researchers that disaggregate socio-economic characteristics find that individual attitudes may be more important than these aggregated measures that serve as a proxy [21,53].

In municipal parks, education level and park age were only occasionally important [21,54]. These aggregated measures are thought to influence vegetation through neighborhood investment, advocacy, and legacy effects [40,44,55]. Individual scale drivers on other land uses are poorly studied.

Developers for all land uses are often motivated by cost and investment decisions [56]. Bulk construction paired with removing existing vegetation is purportedly cheaper, though preserving vegetation may be less expensive in the long run [14].

These management decisions which create vegetation communities and patterns in cities also impact ecosystem function, food webs, and biodiversity [1,2,13,14,57,58]. Different tree and shrub species have different capacity for carbon sequestration [59,60]. Introduced ornamentals generally do not same insect species, or the same biomass or diversity of fauna as native habitat [14,23,61–63]. These changes to habitat quality and quantity also impact higher trophic levels [1,23,64–68]. For the urban matrix to support conservation, decision makers across land uses need to take actions that support locally important vegetation habitat [69,70].

While the drivers and outcomes of decision making are increasingly well studied on residential private property, other land uses have not been given the same attention [71,72]. For example, commercial and industrial land uses are generally included only as independent variables in remote sensing studies of factors influencing percent canopy cover [51,73]. Additionally, research where the unit of analysis is defined by the area of influence of specific decision makers is also needed. Aggregated measures, such as vegetation transects through neighborhoods or canopy cover of a census block, cannot examine specific decision outcomes as they conflate different actors and their motivations and actions, and previous research shows that motivations differ between actors [21,59].

To fill this gap, I examined woody vegetation community composition on office developments in Bellevue and Redmond, Washington, USA. Specifically, I examined 1) tree and shrub communities present on office developments and 2) whether aggregated and parcel specific socio-economic variables or development and landscaping outcomes better explained observed variation in vegetation communities.

I hypothesized that vegetation communities on office developments would be heterogeneous. I also hypothesized that aggregated socio-economic variables found significant in explaining vegetation patterns on residential property would not be significant on office developments [2,34,42], but that parcel scale variables would. Finally, I hypothesized that the outcome of development and landscaping actions would better explain variation in tree and shrub community structure. I found that woody vegetation communities on office developments are heterogenous with distinct community types, and that in contrast with residential property,

development and landscaping actions explain this variability better than socio-economic variables.

## Materials and methods

This study was approved by the University of Washington Human Subjects Division under Determination of Exemption #48246. Field surveys were approved in writing by private property owners or managers of office developments. I did not perform any animal research or collect plant, animal, or other materials from a natural setting.

### Study area and site selection

Redmond (2017 population 64,000) and Bellevue (population 144,000) are located east of Seattle in King County, Washington [74]. Both cities share a similar ecological history, a similar disturbance timeline for logging and agriculture, and have grown considerably since the opening of the Evergreen Point Floating Bridge (SR 520) in 1963. They are at similar elevations (< 160 m) and experience the same climate and similar weather.

The sampling frame was limited to Redmond and Bellevue north of I-90, excluded developments in Bellevue's central business district, and contained parcels defined as office use by the King County Assessor's Office (Fig 2). I grouped adjacent parcels built within three years of one another and with the same owner to create a unit of analysis based on human action not cadastral boundaries. This initial population size was 492 developments.

I used disproportionate stratified random sampling to ensure that my sample included sites across the entire vegetation gradient. I classified the vegetation at each potential study site into categories using visual estimation during site visits in winter 2014 (Fig 3, Table 1). Sites with no vegetation, with wetlands, or those that were currently under construction or undergoing landscape replanting were excluded from the analysis (total 87 sites). The remaining pool of 405 potential sites had no notable hydrological features on site.

I conducted stratified random sampling on sites with High, Medium Canopy, Medium Diverse, Medium, and Low vegetation types. Site selection was restricted based on site area and surrounding impervious surfaces for concurrent studies [75]. Limiting sampling of these extremes reduced my ability to detect community differences along these gradients, though socio-economic variables are not covariate.

I requested property access through three mailings sent to the property owner or manager on file in the King County Assessor's database [76]. I targeted vegetation categories underrepresented in my sample in the second and third mailings. Of 46 mailed requests, 20 (43.5%) received no response or were not deliverable. Of the 26 (56.5%) responses received, 6 (23.1%) of were rejected and 20 (76.9%) were accepted in writing by an individual with authority to do so [76].

Commercial use of sample sites included light industrial, white collar office space, and medical/dental offices. Some sites were fully leased to tenants, while others were either partly or fully owner-occupied. Company size ranged from less than 10 to many thousand employees.

### Independent variables

I examined three categories of independent variables: 1. aggregated and parcel scale socio-economic variables, 2. development and landscaping outcome variables, and 3. ground cover material and maintenance regime variables (Table 2). Aggregated socio-economic variables, sometimes called neighborhood socio-economic variables, reflect group membership of individual landowners aggregated to a common level (in the United States, often block groups). Parcel scale socio-economic variables reflect individual pieces of land, including building age

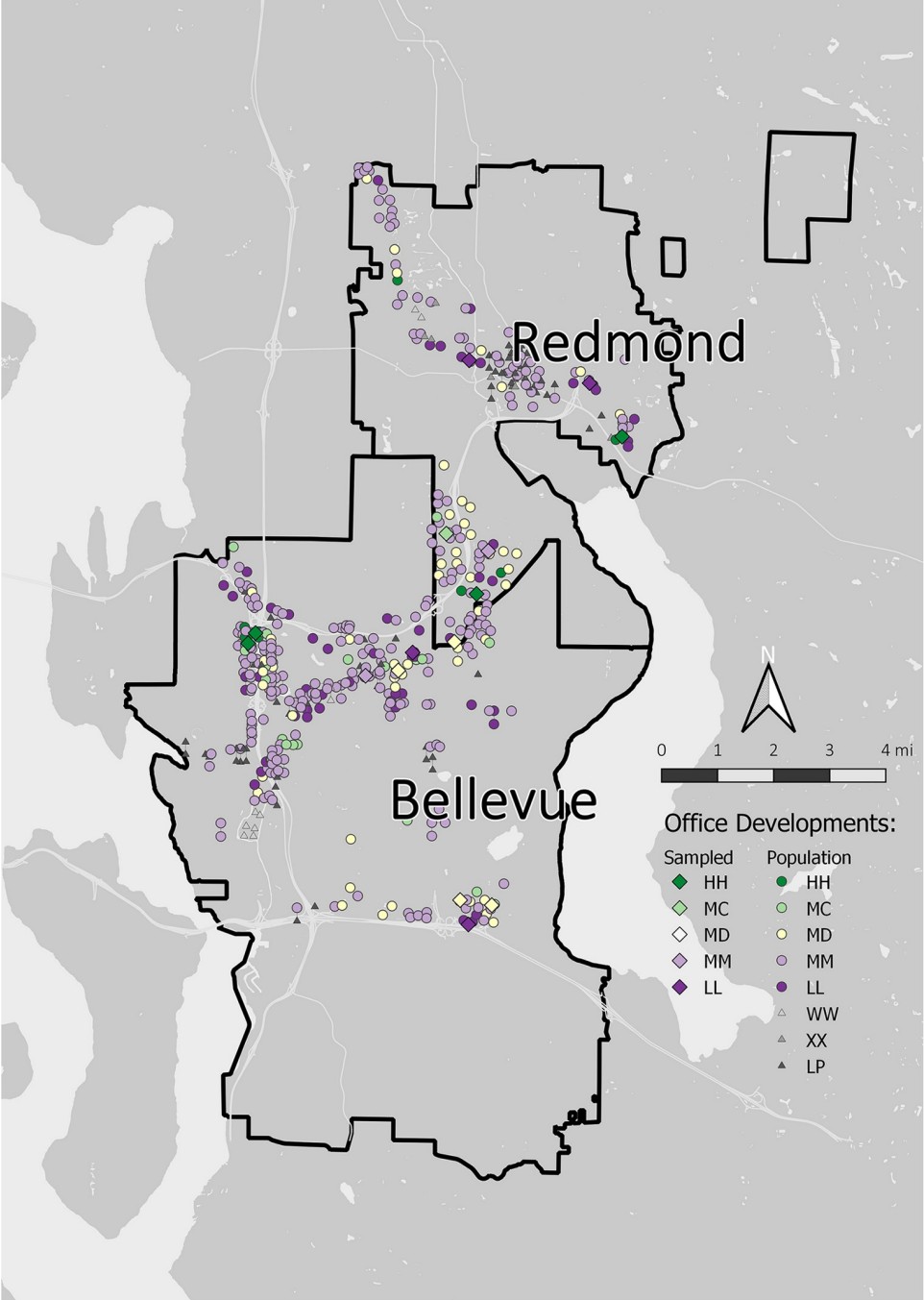

**Fig 2. Map of office development study sites in Redmond and Bellevue, Washington.** The population of office developments with High (HH), Medium Canopy (MC), Medium Diverse (MD), Medium (MM), and Low (LL) vegetation types are represented with colored circles; excluded sites (no vegetation/LP, wetlands/WW, and under construction/XX) are represented with gray triangles. Sampled sites are shown with colored diamonds.

and value. Socio-economic variables were derived from existing databases [77–81]. Variables were chosen based on previous research and analyzed in QGIS 3.2 [42,54,82–85].

I measured the height of dominant native conifers using a Nikon Forestry Pro Laser Range-finder. This is a proxy measure for age as I did not collect tree cores due to liability concerns

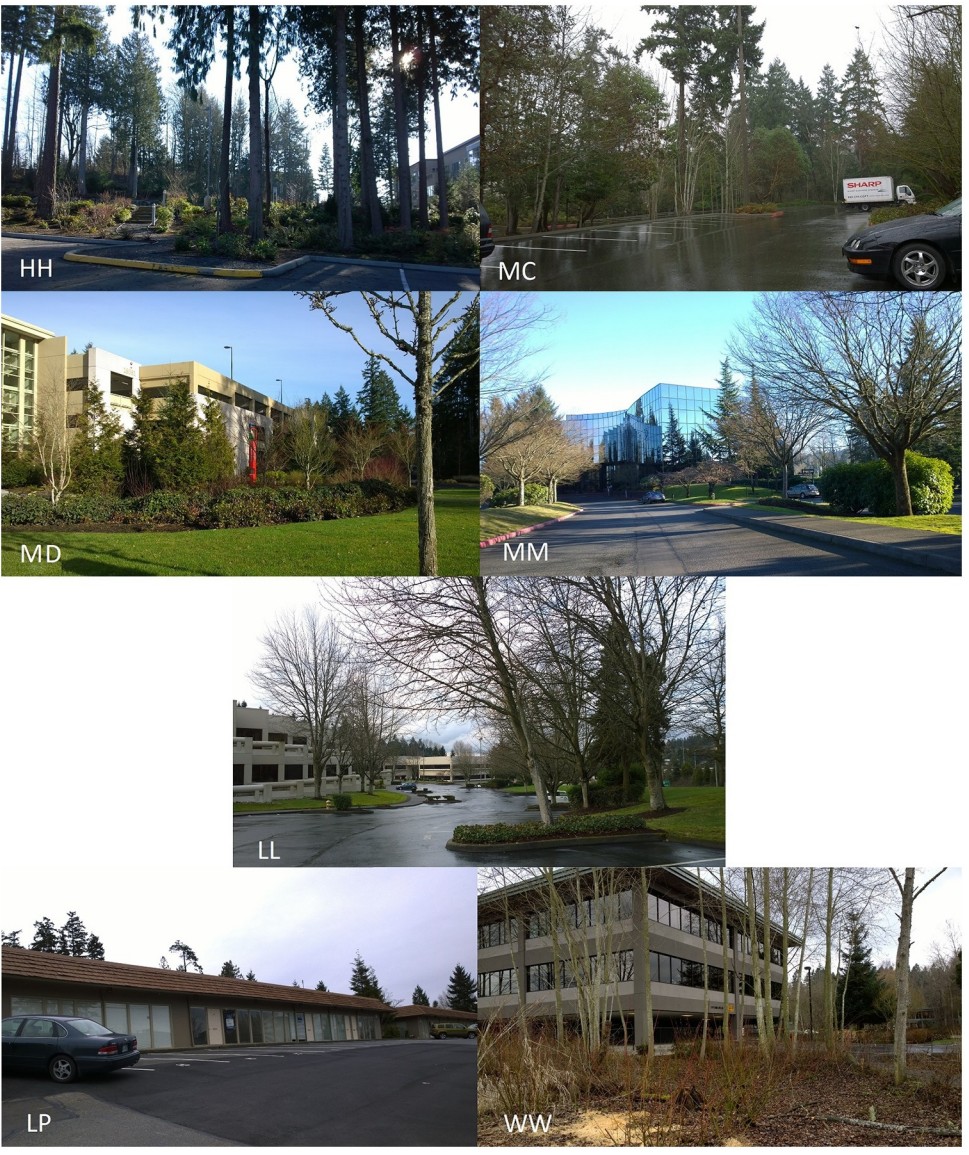

**Fig 3. Examples of each vegetation type.** From top left to bottom right: High (HH); Medium Canopy (MC); Medium Diverse (MD); Medium (MM); Low (LL); no vegetation (LP; excluded); wetlands (WW; excluded).

Table 1. Vegetation type assignment criteria and strata size.

| Vegetation Type | Tree Cover | Shrub Richness | Strata Size | Sampled (n) | Notes |
|---|---|---|---|---|---|
| High | 30% native tree cover | > 5 native shrub genera | 10 | 5 | |
| Medium Canopy | 30% native tree cover | No requirement | 22 | 3 | |
| Medium Diverse | 15% tree cover | > 5 native shrub genera | 53 | 4 | |
| Medium | 15% tree cover | > 5 shrub genera | 264 | 3 | |
| Low | < 10% tree cover | < 5 shrub genera | 56 | 5 | |
| No Vegetation | No trees | No shrubs | 71 | 0 | Excluded from further analysis |
| Wetlands | No requirement | No requirement | 10 | 0 | Excluded from further analysis |
| Under Construction | No requirement | No requirement | 6 | 0 | Excluded from further analysis |

**Table 2. Definition of independent variables used in PERMANOVA and correlation analysis [77–81].**

| | Definition | Data Source | Population | Sample |
|---|---|---|---|---|
| **1. AGGREGATED AND PARCEL SCALE SOCIO-ECONOMIC VARIABLES** | | | | |
| **Area (acre)** | Site area, in acres. | King County Assessor | Range: 0.14–42.51; Mean (SD): 3.61 (5.51) | Range: 0.63–5.39; Mean (SD): 2.57 (1.58) |
| **Town** | Location; Bellevue or Redmond. | King County Assessor | Bellevue: 281 Redmond: 123 | Bellevue: 13 Redmond: 7 |
| **Building Age (in 2017)** | Age of building on site (or mean age for multiple buildings) in 2017. | King County Assessor | Range: 4–99; Mean (SD): 33.2 (11.82) | Range: 9–42; Mean (SD): 32.1 (9.8) |
| **Building Quality** | Categorical 'quality class' assigned to buildings on the site | King County Assessor | Below Average: 11 Average: 146 Average/Good: 96 Good: 120 Good/Excellent: 25 | Below Average: 0 Average: 7 Average/Good: 4 Good: 7 Good/Excellent: 2 |
| **Appraised land value (USD/acre)** | Appraised land value divided by site area. One missing assessed land value was replaced with population median land value. | King County Assessor | Range: 214,673–6,086,305; Mean (SD): 1,845,520 (904,065) | Range: 578,266–3,028,353; Mean (SD): 1,679,110 (623,031) |
| **Impervious w/in 500 m (%)** | Percent impervious surface within 500 m of the site's perimeter. | National Land Cover Database 2011 Percent Developed Imperviousness dataset updated in 2014 | Range: 19.5–81.1; Mean (SD): 55.8 (11.6) | Range: 48.8–67; Mean (SD): 56.8 (6.3) |
| **Median household income (USD)** | The median income of residents for the site's block group | American Community Survey 2014 5-year block group | Range: 42,368–194,107; Mean (SD): 81,408 (24,957) | Range: 42,368–134,643; Mean (SD): 80,478 (22,179) |
| **Foreign-Born (%)** | The percent of residents born outside of the United States for the site's block group. | American Community Survey 2014 5-year block group | Range: 14.6–86.1; Mean (SD): 39 (16.7) | Range: 14.6–86.1; Mean (SD): 40.6 (18.3) |
| **2. DEVELOPMENT AND LANDSCAPING OUTCOME VARIABLES** | | | | |
| **Stands predate development** | Binary variable indicating presence of a cluster of three + trees that predate development. | Site survey | NA | Yes: 12 No: 8 |
| **Median height of dominant conifers (m)** | Median height (m) of five dominant native conifer trees; age proxy. | Site survey | NA | Range: 0–40.6; Mean (SD): 25.8 (13) |
| **Density of native conifers (trees/acre)** | Total density of Douglas-fir, western redcedar, and western hemlock. | Site survey | NA | Range: 0–61.3; Mean (SD): 22.5 (19.3) |
| **3. GROUND COVER MATERIAL AND MAINTENANCE REGIME VARIABLES** | | | | |
| **Ground cover (%)** | Ground cover types on site including lawn, mulch, and impervious surface. | Site survey | NA | Mean (SD) Grass: 7.3 (6.9); Impervious: 66.4 (10.5); Dirt/Litter: 6 (8) |
| **Dead wood (count)** | Total abundance of stumps, logs, and snags on site. | Site survey | NA | Range: 0–40.6; Mean (SD): 25.8 (13) |
| **Irrigation** | Binary variable indicating whether irrigation is used during the summer months. | Interviews and site survey | NA | Yes: 16 No: 3 |
| **Mulch, herbicide, and/ or fertilizer application** | Binary variables (3) indicating whether landscaping crew applies mulch, herbicides, or fertilizers to a site. | Interviews and site survey | NA | Mulch Y/N: 17/3 Herbicide: 13/4 Fertilizer: 15/3 |

Summary statistics for independent variables for both the population of office developments in Redmond and Bellevue and the sample of sites studied (405 and 20 sites, respectively). Median income ($) and proportion foreign born are included to compare patterns in commercial developments with patterns found significant in residential research.

[76]. I used historical records and site construction plans to determine whether each site retained a stand of three adjacent tree predating site development. I used *Pseudotsuga menziesii* (Mirb.) Franco, *Thuja plicata* Donn ex D. Don, and *Tsuga heterophylla* (Raf.) Sarg. counts to calculate native conifer density.

I digitized broad ground cover material classes in QGIS to calculate area [85]. Pervious cover types recorded include dense vegetation, dirt/litter, lawn (turf grass including moss and forb species), gravel, dense ivy, mulch, and water. I used semi-structured interviews of property owners, managers, and landscaping services along with site visits to obtain maintenance regime variables [86,87]. Irrigation, mulching, herbicide, and fertilizer application had only three "no" responses and thus could not be used to draw any well supported conclusions.

## Vegetation data collection

I censused vegetation communities during the summer of 2015, excluding saplings with DBH < 3". Each tree or shrub was identified to species or genus in consultation with experts at the Center for Urban Horticulture at University of Washington [88–90]. Some tree and shrub species were grouped at the genus level due to the abundance of very similar cultivars in the landscaping trade, including *Malus* Mill. [46]. Following previous studies, I grouped conifers under 2 m into a broad class of dwarf conifer species [91]. 10 individual trees (0.506% of total trees) and 14 shrubs (0.174% of total shrubs) could not be identified; these were given a unique identifier code for multivariate community analysis.

I assigned tree and shrub genera to one of three provenance categories—native, non-native, or ambiguous [92,93]. The ambiguous category was used for genera including both native and non-native cultivated species that are difficult to distinguish, and/or frequently interbred and sold as crosses. For example, some *Mahonia* Nutt. species are native (*M. aquifolium* Pursh Nutt. and *M. nervosa* Pursh Nutt.), while others originate in Asia (*Mahonia japonica* Thumb. DC.) and many hybrids are bred and sold by nurseries (e.g. *Mahonia x media* "Charity" Brickell).

## Identifying and describing vegetation clusters on office developments

I standardized tree and shrub abundance data and ground cover area by total site area in acres. This transformation preserves parcel boundaries as the unit of analysis and reflects developer and landowner actions during and following development that determine the amount of impervious surface and pervious area, the number of trees preserved, and the number of trees and shrubs planted. All analyses were performed in R [94].

To delineate vegetation community clusters on office developments, I used a flexible agglomerative nesting function (agnes {vegan}) with a beta of -0.5 to produce an ecologically interpretable dendrogram with minimal chaining [95–99]. Using the resulting groups, I performed indicator species analysis, which assesses the predictive values of species as indicators of the conditions at site groups, using multipatt {indicspecies} [100–102] and a custom wrapper for repetition of the permutation-based function [103]. To examine citywide patterns, I extrapolated cluster membership to the entire population of office developments in the study area using proportions.

I used simple univariate permutational analysis of variance (PERMANOVA) models to test if continuous variables differed between vegetation community cluster groups and Pearson's Chi-squared test to test if categorical variables differed [99,104]. PERMANOVA is a permutation-based implementation of analysis of variance (ANOVA) that avoids assumptions about underlying distributions of community structure and can be used with non-Euclidian distance

matrices [104]. Bartlett tests of homogeneity found no difference between group variances (bartlett.test {stats}).

### Explaining variation in tree and shrub community structure

I analyzed tree and shrub communities separately to determine if the two communities responded differently, and as development and landscaping outcome variables were derived from measurements of the tree community.

I used non-metric multidimensional scaling (NMDS) to evaluate relationships between development and landscaping variables and the tree community [95,99]. NMDS is a rank-based ordination technique that is robust to data without identifiable distributions, can be used with any distance or dissimilarity measure [95]. To determine the relationship between development and landscaping outcome variables and the tree community, I used convex hull plots and fitted environmental vectors [99].

I used PERMANOVA to test relationships between other variables and the tree community and all variables and the shrub community. I used a multi-step approach to avoid transforming independent variables or using ordination to collapse related variables, as these actions make results less interpretable for urban planners and other professionals. I first tested each independent variable with community matrices in a simple multivariate PERMANOVA model. To ensure differences in categorical variables were due to location and not dispersion, I used ANOVA to test for significant differences in dispersion [99]. I then constructed models using all variables with significant pseudo-$F$ values in all possible single and multiple multivariate model combinations. Significance was assessed at the $\alpha \leq 0.05$ level following Holm-Bonferroni correction for multiple comparisons. I used a custom Akaike information criterion with correction (AICc) function based on residual sums of squares to compare models and identify those with the best support [103].

## Results and discussion

### Observed woody vegetation communities

I recorded a total of 1,978 trees and 8,039 shrubs from 52 and 84 taxonomic groups respectively (S1 and S2 Tables). Only *Rhododendron* L. were found on all 20 sites surveyed. Four tree species and nine shrub species were found on more than half of all office developments, with 23 tree species and 30 shrub taxa found only on only one development.

Native tree species accounted for 68.1% of total individuals observed, and three of the top five most abundant species. On average, native species accounted for 63.4% of the trees found on each office development, though sites varied widely with 0%–99% native trees. *Pseudotsuga menziesii* was by far the most abundant tree species, with 37.7% of observed individuals. *Thuja plicata* (12.4%), *Acer macrophyllum* Pursh (11%), *Acer rubrum* L. (6.7%), and *Acer platanoides* L. (5.1%) complete the top five. *Prunus* L. and *Alnus rubra* Bong. were both widespread taxa (found on 12 and 9 sites, respectively) but were never abundant on any one site.

In contrast, native shrub species accounted for only 30.4% individual shrubs observed. On average, native shrubs accounted for 26% of the shrubs observed at each office development, and never more than 63.2% of individual shrubs. The two most abundant shrub species were the native *Gaultheria shallon* Pursh (15.8%), which frequently occurs in low, dense mats, and *Berberis Mahonia gp*. Nutt. (12.5%) which is comprised of native, introduced, and hybrid species. The rest of the top five most abundant shrub species were all non-native, including *Prunus laurocerasus* L. (8.5%), *Rhododendron* (7.6%), and *Cornus sericea* L. (5.2%).

Measures of tree and shrub abundance, density, and diversity varied substantially between sites (Table 3). In general, total species richness and native species richness were positively

**Table 3. Metrics for tree and shrub communities on sampled office developments.**

|  | Minimum | Maximum | Median | Mean | S.D. |
|---|---|---|---|---|---|
| **TREE COMMUNITY** |  |  |  |  |  |
| Tree Abundance | 10 | 240 | 86 | 98.9 | 64.4 |
| Native Tree Abundance | 0 | 230 | 42 | 67.4 | 68.6 |
| Native Conifer Abundance | 0 | 216 | 28 | 49.8 | 57.6 |
| Tree Density | 15.2 | 104.8 | 31.4 | 43.5 | 26.2 |
| Native Tree Density | 0 | 103.6 | 26.9 | 32.9 | 30.5 |
| Native Conifer Density | 0 | 61.3 | 19.7 | 22.5 | 19.3 |
| Tree Species Richness | 3 | 16 | 7 | 8.6 | 3.7 |
| Native Tree Species Richness | 0 | 8 | 4 | 3.9 | 2.3 |
| Tree Shannon Diversity | 0.6 | 2.2 | 1.5 | 1.5 | 0.4 |
| Native Tree Shannon Diversity | 0 | 1.6 | 0.9 | 0.7 | 0.6 |
| Tree Effective Species Richness | 1.9 | 8.7 | 4.7 | 4.8 | 1.9 |
| Native Tree ESR | 1 | 4.7 | 2.5 | 2.4 | 1.2 |
| Tree Sorensen | 0.273 | 1 | 0.667 | 0.665 | 0.16 |
| Tree Arrhenius Model z | 0.348 | 1 | 0.737 | 0.729 | 0.141 |
| **SHRUB COMMUNITY** |  |  |  |  |  |
| Shrub Abundance | 71 | 1789 | 220.5 | 401.9 | 439 |
| Native Shrub Abundance | 0 | 675 | 48.5 | 122 | 195.6 |
| Shrub Density | 39.6 | 404 | 125.7 | 153.1 | 99.7 |
| Native Shrub Density |  |  |  |  |  |
| Shrub Species Richness | 8 | 40 | 18 | 18.1 | 7 |
| Native Shrub Species Richness | 0 | 10 | 4 | 4 | 2.6 |
| Shrub Shannon Diversity | 1.7 | 3 | 2.3 | 2.3 | 0.3 |
| Native Shrub Shannon Diversity | 0 | 1.6 | 1.1 | 0.9 | 0.5 |
| Shrub ESR | 5.7 | 20.6 | 10.1 | 10.5 | 3.5 |
| Native Shrub ESR | 1 | 4.9 | 2.9 | 2.9 | 1.2 |
| Shrub Sorensen | 0.357 | 0.92 | 0.613 | 0.63 | 0.109 |
| Shrub Arrhenius Model z | 0.441 | 0.941 | 0.69 | 0.702 | 0.096 |

H' is Shannon's diversity index [105], effective species richness (ESR) = exp(H') [106], density = individuals per acre.

correlated (Pearson's Correlation for Tree: 0.594; Shrub: 0.545), though four sites with above average species richness had three or fewer native species planted. Remnant large native conifer abundance, primarily *Pseudotsuga menziesii*, greatly contributed to sites with greater tree abundance (Pearson's: 0.83); consequently, Shannon diversity was generally lower on sites with more native trees (Pearson's: -0.407).

Overall, these measures are within the ranges reported by other urban ecology studies, though differences in methodology and particularly the use of small plots [47] and remote sensing [48] in other studies and stratified sampling along a vegetation gradient in this study make comparison more difficult. The most abundant tree species on office developments are similar to those on residential properties in western Washington [51,59]. Measures of diversity were generally lower than residential property [47,54]. Species richness was comparable to other commercial land uses and city parks [47,54] though lower than residential land uses [46,47,52,54]. Measures of beta diversity, suggesting low similarity between locations, were also comparable [46].

## Divergent vegetation groups found on office developments

I identified two groups of tree and shrub vegetation (flexible beta = -0.5; agglomerative coefficients of 0.87 and 0.76 respectively; Table 4). Using indicator species analysis, I identified the Native Tree group (11 sites) as characterized by *Thuja plicata*, *Acer macrophyllum* Pursh, *Arbutus menziesii* Pursh, and *Alnus rubra* Bong, while the Ornamental Tree group (9 sites) is characterized by *Acer rubrum* L. The Native Shrub group (11 sites) is characterized by *Gaultheria shallon* Pursh, *Mahonia gp.* Nutt., *Symphoricarpos* Duham., and *Ribes sanguineum* Pursh, and the Ornamental Shrub group (9 sites) by *Thuja occidentalis* L.

The two groups are distinct in the average density of trees and shrubs per site (Native Tree mean = 58, Ornamental Tree mean = 25.7 with $Pr(>F) = 0.003$; Native Shrub mean = 226.6, Ornamental Shrub mean = 92.9 with $Pr(>F) = 0.001$). The mean median height of dominant native conifers was also significantly different between clusters for trees and shrubs (Native Tree mean = 33.2 m, and Ornamental mean = 16.8 m, with $Pr(>F) = 0.001$; Native Shrub mean = 32.6 m, and Ornamental mean = 20.2 m, with $Pr(>F) = 0.03$). However, there was no difference in area between Native and Ornamental clusters for either trees or shrubs (tree $Pr(>F) = 0.424$; shrub $Pr(>F) = 0.599$). Dead wood abundance was significantly greater on Native Tree sites than Ornamental Tree sites (Native Tree mean = 13.4, and Ornamental mean = 2.4, with $Pr(>F) = 0.019$), but not between shrub groups.

Only impervious surface cover between Native and Ornamental Tree sites differed significantly (Native Tree mean = 60, and Ornamental mean = 70, with $Pr(>F) = 0.007$). No other ground covers differed.

There was also substantial co-occurrence between Native and Ornamental groups. Of the 20 office developments surveyed, nine sites belong to both Native Tree and Shrub community groups, and seven sites belong to both Ornamental Tree and Shrub community groups. This suggests that the sequential decisions made concerning tree preservation, tree plantings, and shrub plantings are related. The observed differences in species composition, between group differences, and high turnover (beta diversity) support the conclusion that woody vegetation communities on office developments are heterogenous.

Native Tree and Shrub communities are more rare than Ornamental Tree and Shrub communities. Extrapolation suggests there are approximately 70 Native Tree and 335 Ornamental Tree developments (17.3%), and 152 Native Shrub and 253 Ornamental Shrub developments (37.5%). The accuracy of these estimates is influenced by the Medium vegetation type, as it is large and proportionally under sampled, and the relatively small sample size.

**Table 4. Rank abundance of tree and shrub taxa for each community group identified by flexible-beta cluster analysis.**

|  | Native Tree Group | Ornamental Tree Group | Native Shrub Group | Ornamental Shrub Group |
|---|---|---|---|---|
| 1. | *Pseudotsuga menziesii** (58.6) | *Pseudotsuga menziesii** (11.2) | *Gaultheria shallon** (106.1) | *Prunus laurocerasus* (57.3) |
| 2. | *Thuja plicata** (20.4) | *Acer rubrum* (10.9) | *Berberis Mahonia* gp. (84) | *Rhododendron* sp. (36.6) |
| 3. | *Acer macrophyllum** (19.4) | *Acer platanoides* (10.4) | *Rhododendron* sp. (25.7) | *Cornus sericea* gp. (23.4) |
| 4. | *Acer rubrum* (3.1) | *Pinus nigra* (8) | *Cornus sericea* gp. (18.9) | *Lonicera pileata* (15.1) |
| 5. | *Alnus rubra** (2.2) | *Callitropsis nootkatensis** (5.4) | *Acer circinatum** (18.3) | *Viburnum davidii* (13.7) |
| 6. | *Arbutus menziesii** (1.7) | *Acer saccharum* (4.8) | *Vaccinium ovatum** (16.1) | *Berberis thunbergii* (13.1) |
| 7. | *Populus tremuloides* (1.5) | *Fraxinus americana* (3.9) | *Prunus laurocerasus* (15.1) | *Gaultheria shallon** (11.1) |
| 8. | *Liquidambar styraciflua* (1.2) | *Prunus* subg. *Cerasus* (3.3) | *Viburnum davidii* (14.1) | *Ilex crenata* (10.1) |
| 9. | *Prunus* subg. *Cerasus* (0.8) | *Thuja plicata** (2.3) | Symphoricarpos sp.* (13) | Ornamental conifer (9.9) |
| 10. | *Callitropsis nootkatensis** (0.7) | *Fraxinus pennsylvanica* (1.9) | *Ribes sanguineum** (12.5) | *Berberis Mahonia* gp. (9.2) |

Asterisk indicates native tree and shrub species. Number in parenthesis is mean abundance of the species in the community group.

## Socio-economic variables poorly explain variation in tree or shrub community composition

Neither aggregated neighborhood scale measures of residential socio-economic status nor parcel scale measures of economic value and the built environment explained variation in tree and shrub community composition on office developments following Holm-Bonferroni correction (S3 Table). For the tree community, median household income was significant only before correction. This largely supports my hypothesis that aggregated socio-economic variables describing neighborhoods are not important for adjacent commercial properties as well [41,42], though additional research is needed.

Theoretically, for residential socio-economic variables to drive vegetation on office developments, the adjacent residential context must influence developer and commercial landowner vegetation choices. Generally, zoning code in Bellevue and Redmond seeks to screen land uses from one another. Owners of office developments are likely signaling to prospective and existing tenants [37,38], in contrast with owners of residential properties, who use vegetation choices to signal to their neighbors of similar socio-economic status [27,35,36]. Other explanations include proximity to desirable amenities–that is, both residential and commercial properties near amenities are appealing to more wealthy neighbors/tenants, the influence of city design review boards, or neighborhood overlay districts.

Studies examining why decision makers on commercial property make planting decisions are fewer in number than residential homeowners, though existing studies provide important early insight. For example, for landscape architects in Toronto factors like site aspect, appearance, and available space rated more highly in species selection than whether species are native or nearby canopy composition [34].

Parcel scale measures of economic value and the built environment were not significant, which fails to support my hypothesis. Previous research found that property value explained variation in the woody vegetation community [51]. Similarly, site age was suggested as a determinant of woody vegetation community composition by studies on residential properties [2,44], landscaping professionals I interviewed, and my examination of contemporaneous landscaping plans filed with the cities of Bellevue and Redmond. Landscaping professionals mentioned trends in plant popularity, including *Pieris japonica* (Thunb.) D. Don ex G. Don in the late 1980s and increasing use of native plants like *Ribes sanguineum* since 2000. Alternative explanations for this finding include that building age is a poor measure for landscaping age due to replanting; an interaction between age and landscaping budget; or that a subset of office developments are planted with in vogue landscape plants, such as the common but under sampled Medium vegetation type.

Differences in study design may also be responsible for these divergent results. Other studies use index response variables and univariate regression [42, 54], measures dependent on effort [54, 107], and plot or transect designs which confound different actors and outcomes [47, 72].

## Development and landscaping outcomes are related to tree and shrub community composition

Multiple variables describing the outcome of development and landscaping actions explain variation in tree and shrub community composition. For the tree community, NMDS with convex hulls and fitted environmental vectors found strong relationships with median dominant native conifer height, native conifer density, the presence of stands predating development, and dead wood abundance (particularly stump abundance, Fig 4). These variables

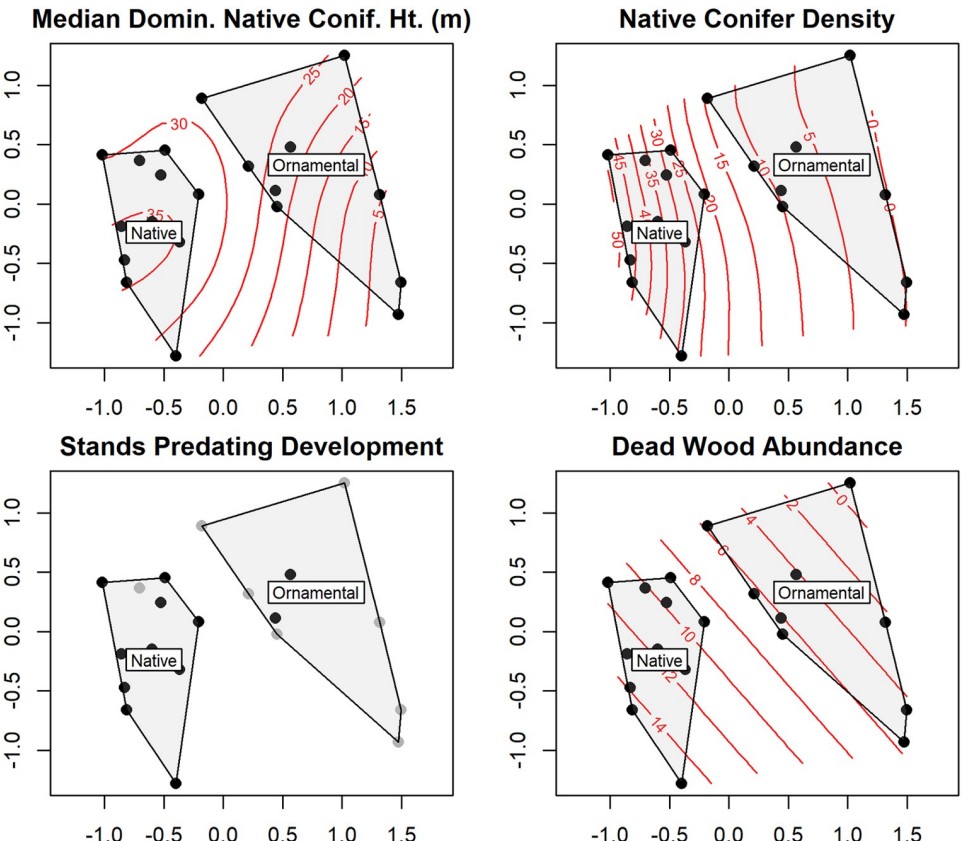

**Fig 4. Two dimensional NMDS representation of tree community composition.** Median dominant native conifer height, native conifer density, and the presence of stands predating development are associated with the first NMDS axis. Dead wood is associated with both axes. Black dots represent sites with stands predating development, gray dots sites without. Ordination has not been rotated prior to plotting.

were also included in the best supported PERMANOVA models for the shrub community (Table 5).

Together, these results support my hypothesis that development and landscaping actions impact vegetation communities on office developments. They also agree with some residential researchers who found that homeowner attitudes and actions were more important than

**Table 5. PERMANOVA model summary comparing multivariate models of shrub community composition.**

| Model | Pseudo-F | p-value | AICc Value | Delta AICc |
|---|---|---|---|---|
| Median height of dominant conifers | 3.08 | 0.001 | 35.1 | 0.00 |
| Tree cluster group (Native v. Ornam.) | 2.86 | 0.001 | 35.4 | 0.21 |
| Native conifer density | 2.82 | 0.003 | 35.4 | 0.25 |
| Tree group + Median height | 2.44 | 0.003 | 36.1 | 0.91 |
| Median height + Native conifer density | 2.27 | 0.002 | 36.4 | 1.22 |
| Stands predate development | 2.26 | 0.011 | 35.9 | 0.79 |
| Median height + Stands predate development | 2.20 | 0.001 | 36.5 | 1.35 |
| Tree group + Native conifer density | 1.87 | 0.014 | 37.1 | 1.97 |
| Tree group + Stands predate development | 1.80 | 0.019 | 37.3 | 2.11 |
| Stands predate development + Native conifer density | 1.80 | 0.018 | 37.3 | 2.11 |

socio-economic descriptors [53]. Together with flexible beta clustering results, this suggests that for each development a suite of decisions is made that results in either retaining more trees and planting native shrubs or retaining fewer trees and planting ornamental trees and shrubs.

However, development and landscaping outcomes are the end point of economic decision making processes poorly studied in urban ecology. Though the coarse socio-economic variables examined here were not significant, developer and landowner motivations and decision making were not considered explicitly, only their outcomes. To reach these end points, developers may consider ease of construction based on site conditions, relative cost of different construction approaches, preferences of the landowner and customer specifications, previous company experience or company aesthetic, and development regulations [13,34,108–110]. The intended audience of prospective and existing tenants may influence both development and landscaping decisions [37,38]. These considerations may influence financing available to developers, financial risk, and the appeal of and thus demand for the completed project [37]. Further, when considering multiple competing options—such as different landscaping choices—developers and landowners may satisfice [111]. That is, they search through alternatives until one meets an acceptability threshold and choose that option.

## Implications for urban habitat quality and quantity

The woody vegetation communities I observed on office developments suggest that integrating habitat conservation during and following development is possible using currently existing developments as models. As local biological communities are largely determined by vegetation, sites with more trees preserved and a greater abundance of native conifers likely provide higher habitat quality and quantity to other organisms [1,2,14,57,58] as native vegetation is more likely to support native insects and native birds than ornamental plantings [23,75,112–116]. One estimate suggests native vegetation volume must be above 70% in order to maintain populations of native insectivorous bird species [116]; sites with high numbers of trees preserved may already hit this target.

We can point to actions and policies more likely to support high quality habitat and benefit other trophic levels, including tree preservation policies, promoting native tree and shrub planting, and removing policy barriers to native vegetation [75,117,118]. However, the motivations driving exemplary adoption of these actions are currently opaque. Anecdotes shared during fieldwork suggest owner-occupied office space, cost, and personal values and connections to nature may be important factors in determining development and landscaping actions, as with homeowners [21,36,119–123].

## Implications for future urban ecology research

Observed within land use heterogeneity and between land use differences in socio-economic variable importance both have implications for urban ecology research. Within land use heterogeneity results in vegetation distributions that are non-normal, with likely kurtosis and heteroscedasticity (e.g. Fig 5). Therefore, choice of sampling design and statistical method can result in inaccurate conclusions, particularly in conjunction with small sample size [124]. Sampling across important gradients, as with vegetation composition in this study, is particularly important. Additionally, potential solutions already extant on the landscape may be overlooked. This provides support for stratified sampling designs, larger sample sizes, and choosing analysis methods robust to broken assumptions of normality of the sampled population [125].

Researchers should choose their sampling strategy carefully based on research questions and the underlying distribution of key variables in urban contexts with long environmental

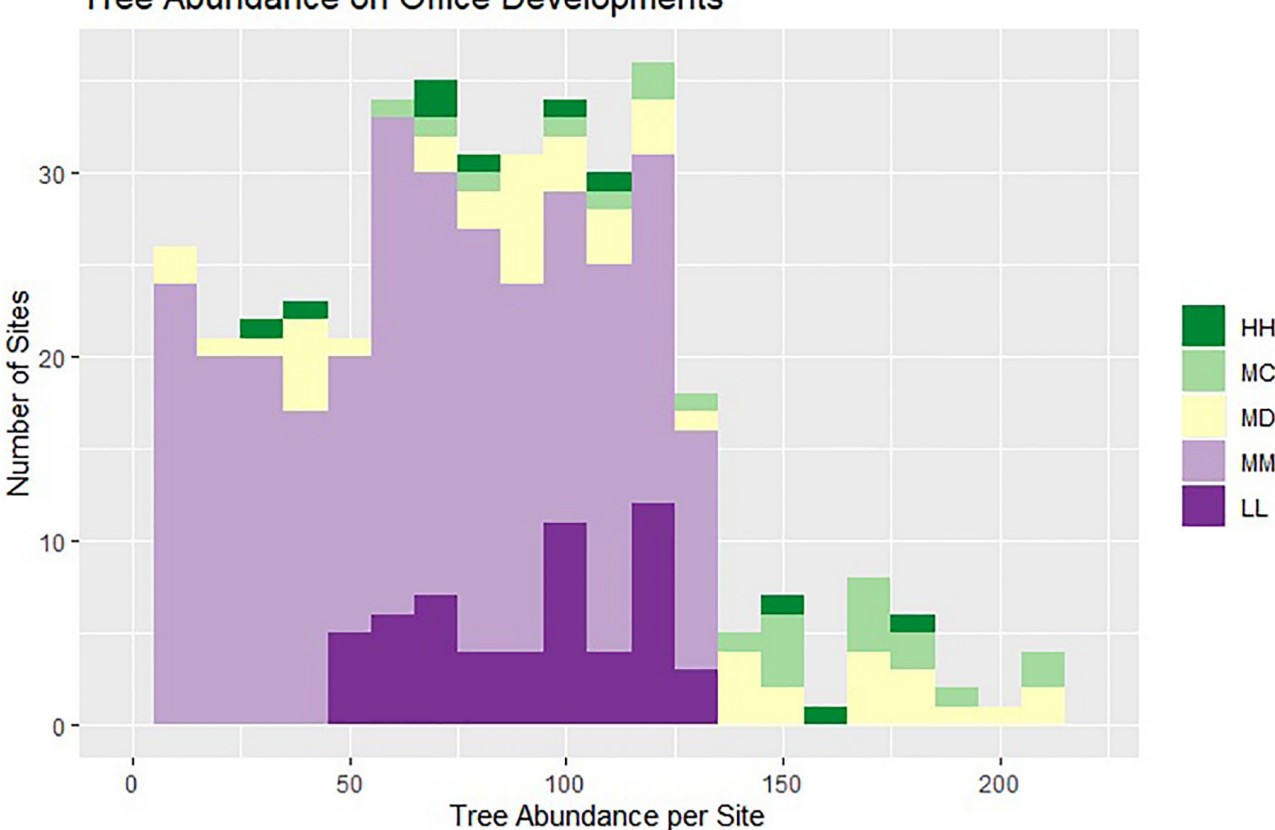

**Fig 5. Hypothesized distribution of the number of trees on office developments based on observed mean and standard deviations for each vegetation class used in sampling (HH, MC, MD, MM, LL).** Note heavy right tail from HH,MC, and MD sites (kurtosis); each vegetation class also has a different variance (heteroscedasticity).

gradients [124,126,127]. If the phenomenon of interest is related to the vegetation community, researchers should attempt to better understand and sample the vegetation gradient (e.g. via stratified sampling) instead of sampling only along a measure of the built gradient [128]. Here, collecting vegetation information for the population of office developments prior to sampling improved sampling design.

Between land use differences in socio-economic variable importance suggests that creating vegetation models of land use within a city is likely inaccurate if all land uses are assumed to respond equivalently to a given variable. Researchers cannot assume that vegetation gradients and socio-economic gradients are parallel; these gradients may also interact resulting in heteroscedasticity. Decision pathways to support carbon sequestration and habitat models need to be constructed based on research for each land use separately.

## Conclusion

Humans control woody vegetation communities in urban ecosystems, influencing ecosystem service provision and habitat quality and quantity. Commercial land uses, including office developments, have largely been overlooked in studies of urban woody vegetation composition and studies examining how these communities are assembled. I filled this gap by examining tree and shrub communities on office developments in Redmond and Bellevue, Washington, USA.

I found that the vegetation communities on these developments are heterogenous, with distinct groups of sites characterized by Native and Ornamental Tree and Shrub vegetation communities. I also found that socioeconomic measures aggregated to the neighborhood scale and at the parcel scale were less important in explaining variation in community composition than variables describing the outcomes of development and landscaping choices.

This research contributes to our understanding of vegetation communities outside of municipal parks and residential land uses. It is also one of few studies that uses site surveys where the unit of measurement is based on how management decisions are made, instead of methods derived from wildlands vegetation research [47,72] or remote sensing [48].

The observed heterogeneity in vegetation communities suggests that different ecosystem functions and habitat quantity and quality are provided on office developments. Greater provision of these functions is possible using currently existing developments as models. Further, within land use heterogeneity suggests that urban ecology research must more carefully consider sampling design, particularly sampling along key environmental gradients. The observed differences in variable importance between office developments and residential land uses suggests that future research and models of the urban ecosystem must account for land uses' different decision pathways.

Going forward, researchers should examine other commercial land uses, commercial land use in additional ecotypes, and particularly the decision pathways followed by actors on commercial and other land uses. This research agrees with other studies suggesting that specific actions are more important than aggregated socio-economic variables [53]. Additional research is needed to link decision makers' personal values and aesthetic preferences, economic motivations, and social norms with tree and shrub community composition on commercial land following work on residential property [27,53]. Needed studies include interviews to better understand tree preservation and planting motivations [34,109]; aesthetic preference studies as on residential developments [129,130]; and tracing decision making pathways based on previous land use [131]. A better understanding of these processes may improve habitat quality and quantity on commercial property [132]. Finally, research is also needed to determine if vegetation inequity observed on residential properties [39] is perpetuated on commercial properties. The No Vegetation type excluded from analysis here was often adjacent to retail use, where worker compensation is generally less than in medical/dental, software, and other white collar jobs in office developments.

## Supporting information

**S1 Table. All trees observed in site surveys.** Abundance is count of individuals belonging to each taxonomic group. Ambiguous indicate both native, non-native, and hybrids used in horticulture.
(DOCX)

**S2 Table. All shrubs observed in site surveys.** Abundance is count of individuals belonging to each taxonomic group. Ambiguous indicate both native, non-native, and hybrids used in horticulture.
(DOCX)

**S3 Table. All simple multivariate PERMANOVA results for tree and shrub communities.**
(DOCX)

## Acknowledgments

This research would not have been possible without the cooperation of the office development landowners and managers who allowed me access to their property. My thanks also to staff at the University of Washington Center for Urban Horticulture for assistance with shrub identification, and to members of the Urban Ecology Research Lab, Jon Bakker, Ken Yocom, Gordon Bradley, Martha Groom, Marina Alberti, and John Owens for their scientific guidance.

## Author Contributions

**Conceptualization:** Karen Dyson.

**Data curation:** Karen Dyson.

**Formal analysis:** Karen Dyson.

**Investigation:** Karen Dyson.

**Methodology:** Karen Dyson.

**Project administration:** Karen Dyson.

**Writing – original draft:** Karen Dyson.

**Writing – review & editing:** Karen Dyson.

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
