## [Decision Letter · Decision Letter 0]

18 Jul 2019

PONE-D-19-17662

Heterogeneity within and between land uses: commercial office vegetation is determined by development and landscaping decisions

PLOS ONE

Dear Dr. Dyson,

Thank you for submitting your manuscript to PLOS ONE. After careful consideration, we feel that it has merit but does not fully meet PLOS ONE’s publication criteria as it currently stands. Therefore, we invite you to submit a revised version of the manuscript that addresses the points raised during the review process.

We would appreciate receiving your revised manuscript by Sep 01 2019 11:59PM. To enhance the reproducibility of your results, we recommend that if applicable you deposit your laboratory protocols in protocols.io, where a protocol can be assigned its own identifier (DOI) such that it can be cited independently in the future. For instructions see: http://journals.plos.org/plosone/s/submission-guidelines#loc-laboratory-protocols

We look forward to receiving your revised manuscript.

Kind regards,

Zhihua Wang, Ph.D.

Academic Editor

PLOS ONE

Journal Requirements:

3. Please ensure that you refer to Figure 1 in your text as, if accepted, production will need this reference to link the reader to the figure.

4. We note you have included a table to which you do not refer in the text of your manuscript. Please ensure that you refer to Table 2 in your text; if accepted, production will need this reference to link the reader to the Table.

Reviewers' comments:

Reviewer's Responses to Questions

**Comments to the Author**

1. Is the manuscript technically sound, and do the data support the conclusions?

Reviewer #1: Yes

Reviewer #2: Yes

2. Has the statistical analysis been performed appropriately and rigorously? 

Reviewer #1: I Don't Know

Reviewer #2: No

3. Have the authors made all data underlying the findings in their manuscript fully available?

Reviewer #1: Yes

Reviewer #2: Yes

4. Is the manuscript presented in an intelligible fashion and written in standard English?

Reviewer #1: Yes

Reviewer #2: Yes

5. Review Comments to the Author

Reviewer #1: This manuscript attempts to fill the identified gap in determinants of vegetation communities within commercial and industrial land uses with a focus on office developments. The paper is generally well written and easy to follow. Also the provided data is sufficient to meet the requirement of PlosONE.

I only have the following suggestions to improve the presentation of two figures:

1. Figure 2: it would be good to colour each vegetation type for better presentation of the heterogeneity; meanwhile the sites sampled can be indicated with symbols different from circles.

2. Figure 5: I doubt a line plot would hinder the interpretation of the underlying distribution: a scatter plot with vegetation types coloured might be more informative.

Reviewer #2: The manuscript “Heterogeneity within and between land uses: commercial office vegetation is determined by development and landscaping decisions” by Karen Dyson aims to investigate the determinant factors of vegetation distribution on commercial office developments through statistical analysis based on the site surveys conducted in Redmond and Bellevue, Washington, USA. The author has collected and analyzed a lot of data on commercial office developments, while others have been mainly focused on residential areas. This is the highlighted novelty of the work, which made it worthwhile to publish after extra revisions and clarifications.

First of all, the title of this manuscript is too long without clear representation of the context. I see that the author tried to link the current findings on commercial office developments to the previous findings on residential areas. Specifically, the key factor of residential area is the socio-economic factor, while the key factors of commercial area are the development & landscaping decisions. However, the manuscript is mainly describing and discussing the data collected in commercial office development areas. Therefore, the title “Heterogeneity within and between land uses” is not appropriate. I would argue that this is only heterogeneity within the commercial land use.

The methodology is not clearly described. For example, the main statistical model PERMANOVA and the non-metric multidimensional scaling (NMDS) method are not well explained before using. A lot of abbreviations like ANOVA and AICc function are not described.

In the section of independent variables, an overview of independent variables should be presented at first. For example, you could mention at the beginning that three categories of independent variables are collected, including 1. aggregated and parcel level socio-economic variables, 2. development and landscaping outcome variables, 3. ground cover material and maintenance regime variables.

Can you define aggregated level and parcel level?

Line 49. What is a Puget Sound Region?

Line 214, what is "agnes" function?

Line 419, you mentioned that "the choice of sampling design and statistical method can result in inaccurate conclusions". Then can you explain how you chose the sampling design and statistical method in this manuscript?

Fig 1 & 4 are not referred anywhere in the context.

Fig 2. Population legend, please add the abbreviations HH for High, MC for Medium Canopy, etc. here since the abbreviations are displayed in the figure.

Some grammar errors exist.

6. PLOS authors have the option to publish the peer review history of their article (what does this mean?). If published, this will include your full peer review and any attached files.

Reviewer #1: No

Reviewer #2: No

---

## [Author Response · Author response to Decision Letter 0]

20 Aug 2019

Reviewer 1:

1. Figure 2: it would be good to colour each vegetation type for better presentation of the heterogeneity; meanwhile the sites sampled can be indicated with symbols different from circles.

2. Figure 5: I doubt a line plot would hinder the interpretation of the underlying distribution: a scatter plot with vegetation types coloured might be more informative.

• I have revised Figure 2 so that each vegetation type is a different color, using a color-blind and printer-friendly color scheme. I have also used different shapes, as suggested.

• I have also revised Figure 5 so that each vegetation type is distinct, using the same color scheme as Figure 2. I used a stacked histogram to show the distribution and each vegetation types’ contribution.

Reviewer 2:

First of all, the title of this manuscript is too long without clear representation of the context. I see that the author tried to link the current findings on commercial office developments to the previous findings on residential areas. Specifically, the key factor of residential area is the socio-economic factor, while the key factors of commercial area are the development & landscaping decisions. However, the manuscript is mainly describing and discussing the data collected in commercial office development areas. Therefore, the title “Heterogeneity within and between land uses” is not appropriate. I would argue that this is only heterogeneity within the commercial land use.

The methodology is not clearly described. For example, the main statistical model PERMANOVA and the non-metric multidimensional scaling (NMDS) method are not well explained before using. A lot of abbreviations like ANOVA and AICc function are not described.

In the section of independent variables, an overview of independent variables should be presented at first. For example, you could mention at the beginning that three categories of independent variables are collected, including 1. aggregated and parcel level socio-economic variables, 2. development and landscaping outcome variables, 3. ground cover material and maintenance regime variables.

Can you define aggregated level and parcel level?

Line 49. What is a Puget Sound Region?

Line 214, what is "agnes" function?

Line 419, you mentioned that "the choice of sampling design and statistical method can result in inaccurate conclusions". Then can you explain how you chose the sampling design and statistical method in this manuscript?

Fig 1 & 4 are not referred anywhere in the context.

Fig 2. Population legend, please add the abbreviations HH for High, MC for Medium Canopy, etc. here since the abbreviations are displayed in the figure.

Some grammar errors exist.

• I have changed the title to reflect my two key conclusions, specifically heterogeneity within land use and importance of development and landscaping decision over socio-economic factors, as noted by reviewer. I have eliminated reference to heterogeneity between land uses, as suggested.

• I have revised the methods section for added clarity, including general language revisions, adding un-abbreviated names for ANOVA and AICc, adding an explanation of PERMANOVA, and adding an explanation of NMDS.

• I added additional text to the beginning of the ‘Independent variables’ section for clarity. I also introduce Table 2, which presents an overview of independent variables.

• For additional clarity, I added the definition of aggregated scale socio-economic variables to the start of independent variable section (in addition to existing discussion in the Introduction) and revised language throughout the manuscript. I also added a definition of parcel level to the independent variable section.

• Line 49: I added additional description of the Puget Sound region.

• Line 214: I added clarifying text about the agnes function

• Line 419: I added multiple sentences clarifying the need to sample across key gradients to obtain accurate results, as I did here.

• I added a reference to Fig 1. Fig 4 was referenced in line 367.

• I added the abbreviations for vegetation types to the caption of Fig 2 to aid with interpretation, as helpfully suggested.

• I found and corrected multiple grammar errors. 

• Reviewed style templates and altered section headings, placement of Fig and Table captions, and file names in response.

• Added references to Fig 1 and Table 2.

---

## [Editor Report · Decision Letter 1]

22 Aug 2019

Vegetation communities on commercial developments are heterogenous and determined by development and landscaping decisions, not socioeconomics

PONE-D-19-17662R1

Dear Dr. Dyson,

We are pleased to inform you that your manuscript has been judged scientifically suitable for publication and will be formally accepted for publication once it complies with all outstanding technical requirements.

With kind regards,

Zhihua Wang, Ph.D.

Academic Editor

PLOS ONE

Additional Editor Comments (optional):

The authors have adequately addressed both reviewers' comments.
---

## [Editor Report · Acceptance letter]

27 Aug 2019

PONE-D-19-17662R1 

Vegetation communities on commercial developments are heterogenous and determined by development and landscaping decisions, not socioeconomics 

Dear Dr. Dyson:

I am pleased to inform you that your manuscript has been deemed suitable for publication in PLOS ONE. Congratulations! Your manuscript is now with our production department. 

With kind regards,

on behalf of

Dr. Zhihua Wang 

Academic Editor

PLOS ONE